# The *Drosophila FUS* ortholog *cabeza* promotes adult founder myoblast selection by Xrp1-dependent regulation of FGF signaling

**Marica Catinozzi**[1,2,3], **Moushami Mallik**[1,2,3], **Marie Frickenhaus**[2,3], **Marije Been**[1], **Céline Sijlmans**[1], **Divita Kulshrestha**[1,2,3], **Ioannis Alexopoulos**[4], **Manuela Weitkunat**[5], **Frank Schnorrer**[5,6], **Erik Storkebaum**[1,2,3]*

1 Department of Molecular Neurobiology, Donders Institute for Brain, Cognition and Behaviour and Faculty of Science, Radboud University, Nijmegen, Netherlands, 2 Molecular Neurogenetics Laboratory, Max Planck Institute for Molecular Biomedicine, Muenster, Germany, 3 Faculty of Medicine, University of Muenster, Muenster, Germany, 4 General Instruments Department, Faculty of Science, Radboud University, Nijmegen, Netherlands, 5 Muscle Dynamics Group, Max Planck Institute of Biochemistry, Martinsried, Germany, 6 Aix Marseille University, CNRS, IBDM, Marseille, France

* e.storkebaum@donders.ru.nl

**Data Availability Statement:** All relevant data are within the manuscript and its Supporting Information files.

## Abstract

The number of adult myofibers in *Drosophila* is determined by the number of founder myoblasts selected from a myoblast pool, a process governed by fibroblast growth factor (FGF) signaling. Here, we show that loss of *cabeza* (*caz*) function results in a reduced number of adult founder myoblasts, leading to a reduced number and misorientation of adult dorsal abdominal muscles. Genetic experiments revealed that loss of *caz* function in both adult myoblasts and neurons contributes to *caz* mutant muscle phenotypes. Selective overexpression of the FGF receptor Htl or the FGF receptor-specific signaling molecule Stumps in adult myoblasts partially rescued *caz* mutant muscle phenotypes, and Stumps levels were reduced in *caz* mutant founder myoblasts, indicating FGF pathway deregulation. In both adult myoblasts and neurons, *caz* mutant muscle phenotypes were mediated by increased expression levels of Xrp1, a DNA-binding protein involved in gene expression regulation. Xrp1-induced phenotypes were dependent on the DNA-binding capacity of its AT-hook motif, and increased Xrp1 levels in founder myoblasts reduced Stumps expression. Thus, control of Xrp1 expression by Caz is required for regulation of Stumps expression in founder myoblasts, resulting in correct founder myoblast selection.

## Author summary

Skeletal muscles mediate movement, and therefore, proper structure and function of skeletal muscles is required for respiration, locomotion, and posture. Adult muscles arise from fusion of muscle precursor cells during development. In the fruit fly *Drosophila melanogaster*, muscle precursor cells come in two flavors: founder cells and fusion-competent cells. The number of founder cells selected during development corresponds to the number of adult muscles formed. Here, we report that inactivation of the *Drosophila caz* gene

**Funding:** This work was supported by funding from the Max Planck Society (to E.S. and F.S.), the Frick Foundation for ALS Research (to E.S.), the Muscular Dystrophy Association (MDA, to E.S.), the EU Joint Programme – Neurodegenerative Disease Research (JPND; grant numbers ZonMW 733051075 (TransNeuro) and ZonMW 733051073 (LocalMND), to E.S.) and an ERC consolidator grant (ERC-2017-COG 770244, to E.S.). M.M. was supported by a Humboldt Research Fellowship and M.F. by a fellowship from the CEDAD graduate school. Stocks were obtained from the Bloomington Drosophila Stock Center (NIH P40OD018537). The funders had no role in study design, data collection and analysis, decision to publish, or preparation of the manuscript.

**Competing interests:** The authors have declared that no competing interests exist.

results in muscle developmental defects. Loss of *caz* function in both muscle precursor cells and the nerve cells that innervate muscles contributes to the muscle developmental defect. At the molecular level, loss of *caz* function leads to increased levels of Xrp1. Xrp1 regulates the expression of many other genes, including genes that produce components of the FGF signaling pathway, which is known to be involved in founder cell selection. In all, we uncovered a novel molecular mechanism that regulates founder cell selection during muscle development.

## Introduction

In *Drosophila*, two phases of somatic muscle development can be distinguished: during embryonic development, a set of muscles is generated that will mediate larval movement, while during metamorphosis, adult muscles are generated that allow for adult movement, including eclosion from the pupal case, locomotion and flight. In the first 24 h after puparium formation (APF), nearly all larval muscles are histolysed and removed by phagocytosis [1]. Larval motor neurons persist through metamorphosis, but their processes are substantially remodeled to innervate the adult muscles [2]. The latter are formed from undifferentiated myoblasts originating from mesodermal precursor cells that were set aside during embryonic development. These cells continue to express Twist in the late embryo and proliferate during larval life [3]. During early pupal stages, proliferation continues and myoblasts spread out by migrating along the peripheral nerves [1]. Around 24 h APF, a subset of adult myoblasts is selected to become founder myoblasts, while the remaining cells become fusion-competent myoblasts which later fuse with founder myoblasts to form the characteristic multinucleate adult myotubes [4].

Founder cell selection is accompanied by a decline in Twist expression and induction of *dumbfounded* (*duf*, also known as *kirre*) expression [4]. Intriguingly, while the selection of founder myoblasts during embryonic development involves Notch-mediated lateral inhibition [5], adult founder myoblast selection is mediated by FGF signaling and independent of Notch signaling [4, 6]. Indeed, manipulation of the expression level of the FGF receptor Heartless (Htl) in adult myoblasts modulates the number of adult founder myoblasts selected [6]. While Htl is expressed in both founder and fusion-competent myoblasts, Stumps (also known as Dof or Hbr), an FGF receptor-specific signaling molecule downstream of Htl, is selectively expressed in founder myoblasts from 24 h APF onwards [6]. Thus, restriction of Stumps expression to founder myoblasts may be a key event in founder myoblast selection. The molecular mechanism that mediates restricted Stumps expression in founder myoblasts is currently elusive.

Here, we identify a novel role for *cabeza* (*caz*) in adult muscle development. Caz is the single *Drosophila* orthologue of FUS, EWSR1 and TAF15, three highly homologous proteins that constitute the FET protein family in humans [7]. The FET proteins are DNA- and RNA-binding proteins involved in gene expression regulation, including transcription, mRNA splicing and mRNA subcellular localization [7]. Each of the three FET proteins has been implicated in the pathogenesis of the motor neurodegenerative disorder amyotrophic lateral sclerosis (ALS) and frontotemporal dementia (FTD) [8]. We previously reported that loss of *caz* function results in failure of pharate adult flies to eclose from the pupal case due to motor weakness. This is at least in part mediated by loss of neuronal *caz* function, as selective reintroduction of Caz in neurons was sufficient to rescue the *caz* mutant eclosion defect. Moreover, selective inactivation of *caz* in neurons was sufficient to induce an adult eclosion defect, albeit with a

fraction of adult escaper flies that display dramatic motor performance deficits and reduced life span [9]. The fact that selective *caz* inactivation in neurons induces an adult eclosion defect that is not as severe as *caz*$^{KO}$ animals suggests that loss of *caz* function in other cell types may contribute to the *caz* mutant adult eclosion defect. We considered the adult abdominal muscles (Fig 1A) as a likely candidate, given their known role in mediating adult eclosion.

We therefore investigated the role of *caz* function in adult abdominal muscle development. Loss of *caz* function resulted in a significant reduction of the number of adult founder myoblasts and dorsal abdominal muscles (DAMs), as well as DAM misorientation. Interestingly, loss of *caz* function in both adult myoblasts and neurons contributed to the deranged development of adult abdominal muscles. Increased expression of Htl or Stumps in adult myoblasts significantly rescued *caz* mutant muscle phenotypes, indicating that impaired FGF signaling contributes to these phenotypes. Consistently, Stumps levels were significantly reduced in *caz* mutant founder myoblasts. It was previously reported that neuronal dysfunction in *caz* mutants is mediated by increased expression of Xrp1, a DNA-binding protein involved in gene expression regulation [10]. Here, we could show that also the newly-discovered *caz* mutant muscle phenotypes are caused by increased Xrp1 expression levels in both adult myoblasts and neurons. Muscle phenotypes induced by increased Xrp1 expression were dependent on the DNA-binding capacity of its AT-hook motif, and Xrp1 overexpression suppressed Stumps expression in adult founder myoblasts. In summary, we uncovered a novel mechanism for regulation of Stumps expression in adult founder myoblasts: Caz controls Xrp1 expression, thus permitting increased Stumps levels in founder myoblasts and correct founder myoblast selection.

## Results

### Loss of *caz* function deranges adult abdominal muscle development

As proper abdominal muscle function is required for eclosion of adult *Drosophila* from the pupal case, we decided to investigate adult abdominal muscle development in two independent *caz* loss-of-function lines (*caz*$^2$ and *caz*$^{KO}$) [9]. We generated *caz* mutants in which the *Myosin heavy chain* (*Mhc*) gene is GFP-tagged to label muscle fibers [11–13]. *In vivo* confocal imaging of dorsal abdominal muscles (DAMs) 96 h APF revealed a reduced number of DAMs in both *caz*$^2$ and *caz*$^{KO}$ (Fig 1B). Quantification of DAM number in abdominal hemisegments 3 and 4 revealed a significant reduction by ~30–40% in both body segments of the two *caz* mutant lines (Fig 1C and 1D). In addition, DAMs were frequently misoriented in *caz* mutant animals (arrowheads in Fig 1B; quantification in Fig 1E and 1F). These abdominal muscle defects are independent of *Mhc-GFP* expression, as immunostaining of late pupal filets for actin confirmed the profound muscle developmental defects, with both reduced number and misorientation of DAMs in *caz* mutants that did not carry additional transgenes (Fig 2A and 2B; S1A Fig). Furthermore, *caz* mutant muscle defects cannot be attributed to a delay in pupal development, as muscle defects were observed both in age-matched (100 h APF) and stage-matched pupae (based on outer appearance [14], S1A and S1B Fig). Importantly, muscle defects were not confined to DAMs, but also observed in other abdominal muscle groups, including ventral abdominal muscles (Fig 2N and 2R). Of note, the larval muscle pattern and number was not affected by loss of *caz* function (S2 Fig), indicating that *caz* mutant muscle defects were confined to adult muscle development, while embryonic muscle development occurred normally.

These muscle morphological defects did not appear secondary to defective innervation by motor neurons, as the number of motor neurons in the third instar larval ventral cord was not altered in *caz*$^2$ animals (Fig 2I and 2J). The fact that *caz* mutant larvae enter metamorphosis with a normal number of motor neurons is relevant for pupal development, as the larval

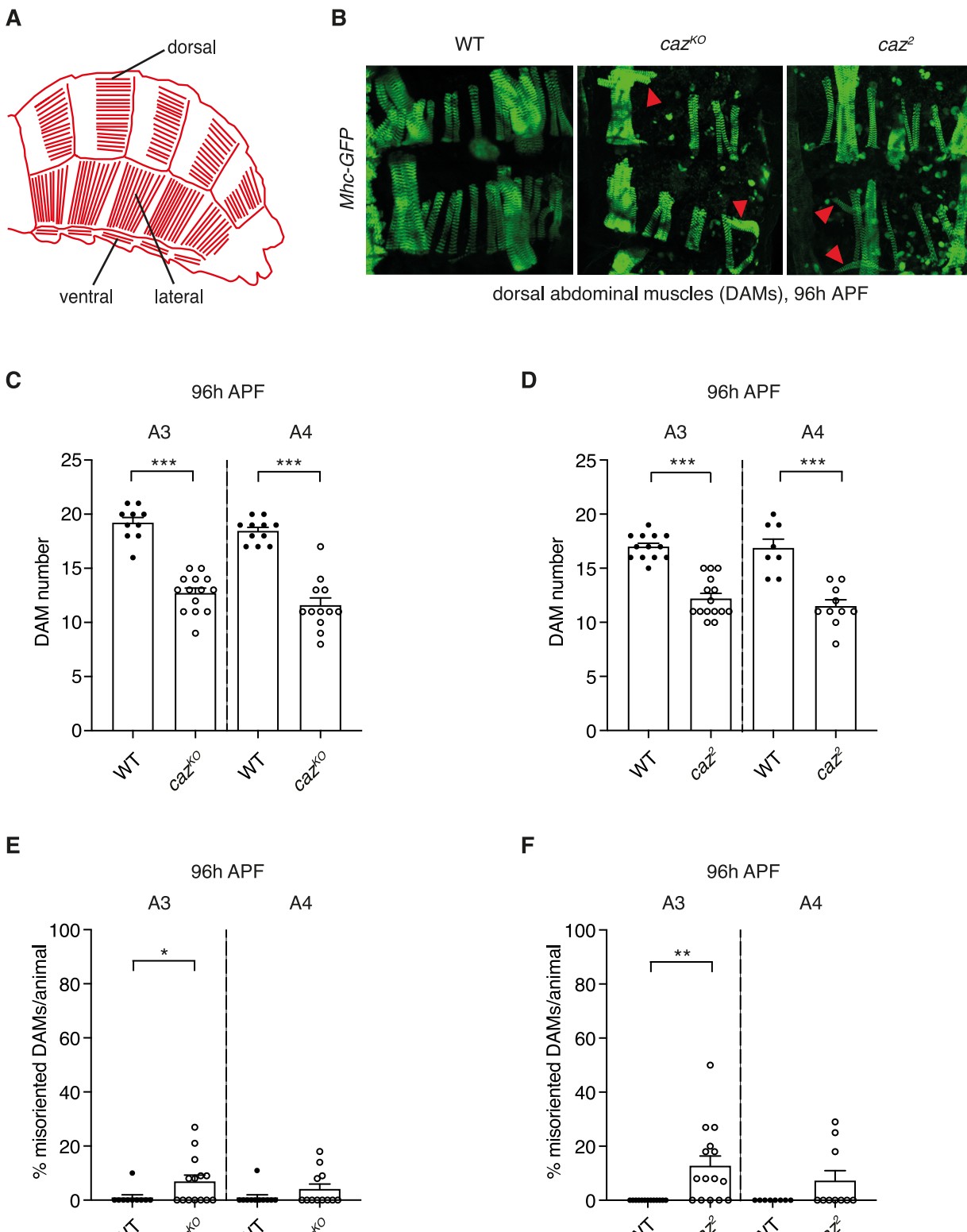

**Fig 1. Loss of *caz* function deranges adult abdominal muscle development. A,** Schematic of adult abdominal muscle architecture in *Drosophila*. Dorsal, lateral and ventral abdominal muscles are indicated. **B,** A transgenic line, in which Myosin heavy chain is GFP-tagged (*Mhc-GFP*), allowed visualization of dorsal abdominal muscles (DAMs) at 96 h APF in wild type (WT), *caz^{KO}*, and *caz^2* animals. Red arrowheads indicate misoriented DAMs. Scale bar: 50μm. **C,D,** Quantification of DAM number in abdominal segments 3 (A3) and 4 (A4) of *caz^{KO}* (C) and *caz^2* (D) pupae at 96 h APF, as compared to WT. Mann Whitney test (*C*), t-test with Welch correction (*D*); ***p<0.0005; n = 10 WT versus 14 *caz^{KO}* (A3), 11 WT versus 12 *caz^{KO}* (A4), 13 WT versus 15 *caz^2* (A3), 8 WT versus 10 *caz^2* (A4). Average ± SEM. **E,F,** Percentage of misoriented DAMs per animal in segments A3 and A4 of *caz^{KO}* (E) and *caz^2* (F) pupae at 96 h APF, as compared to WT. One sample Wilcoxon signed rank test; *p<0.05, **p<0.005; n = 10 WT versus 14 *caz^{KO}* (A3), 11 WT versus 12 *caz^{KO}* (A4), 13 WT versus 15 *caz^2* (A3), 8 WT versus 10 *caz^2* (A4). Average ± SEM.

motor neurons persist to adulthood, in contrast to the majority of larval muscles, which degenerate during metamorphosis [1, 2]. Furthermore, staining of abdominal filets of late pupae (100 h APF) for horseradish peroxidase (HRP), Discs large 1 (Dlg1) and actin revealed normal innervation of both dorsal and ventral abdominal muscles in *caz* knock-out animals (Fig 2A–2H and 2K–2R). This was confirmed by quantitative analysis of NMJ morphology on DAMs of WT and *caz^{KO}* pupae 96 h APF, which revealed that both NMJ area and bouton number were not altered in *caz^{KO}* (Fig 2S and 2T). Thus, loss of *caz* function induces muscle developmental defects, which are not attributable to defective motor innervation.

## Reduced number of founder myoblasts in *caz* mutant animals

As the number of adult muscles, including DAMs, is dependent on the number of founder myoblasts that are selected during early pupal stages (~24 h APF), we investigated whether *caz* is expressed in founder myoblasts, and whether the number of founder myoblasts in the dorsal abdomen was reduced in *caz* mutants. To identify founder myoblasts, a *duf-lacZ* transgenic line was used in which β-galactosidase is selectively expressed in founder myoblasts under the control of the *duf/kirre* promoter [4]. Double immunostaining for Caz and β-galactosidase revealed that Caz is expressed in founder myoblasts (Fig 3A), as well as other cells, likely including fusion-competent myoblasts, consistent with the previously reported broad expression pattern of *caz* [9]. Importantly, the number of founder myoblasts was significantly reduced in *caz* mutant pupae as compared to wild type (WT) control (Fig 3B and 3C). Thus, the reduced number of DAMs in *caz* mutant late pupae is at least in part attributable to a reduced number of founder myoblasts 24 h APF.

## Loss of *caz* function in adult myoblasts contributes to *caz* mutant muscle defects

Our findings suggested that the *caz* mutant muscle defects may be caused by loss of *caz* function in founder myoblasts. In line with this possibility, selective expression of Caz in adult myoblasts (*1151-GAL4*) of *caz* mutants rescued their reduced number of founder myoblasts to a level that was not statistically different from control genotypes (Fig 3D). Consistently, selective Caz expression in adult myoblasts also substantially rescued the reduced DAM number and DAM misorientation of *caz^{KO}* pupae 96 h APF (Fig 3E and 3F). We further examined the effect of selective inactivation of *caz* in adult myoblasts by crossing animals that selectively express FLP recombinase in these cells (*1151-GAL4>UAS-FLP*) to conditional *caz* knock-out animals in which the *caz* gene is flanked by FRT sites (*caz^{FRT}*) [9]. Adult myoblast-selective *caz* inactivation resulted in a significantly reduced number of founder myoblasts at 24 h APF (Fig 3G) and a reduced DAM number and DAM misorientation at 96 h APF (Fig 3H and 3I; representative images in S3A Fig), albeit these muscle developmental defects appeared slightly less pronounced than in full-body *caz* knock-out animals. Together, these data show that Caz has a cell-autonomous function in adult myoblasts to promote founder myoblast selection and to ensure proper DAM number and orientation.

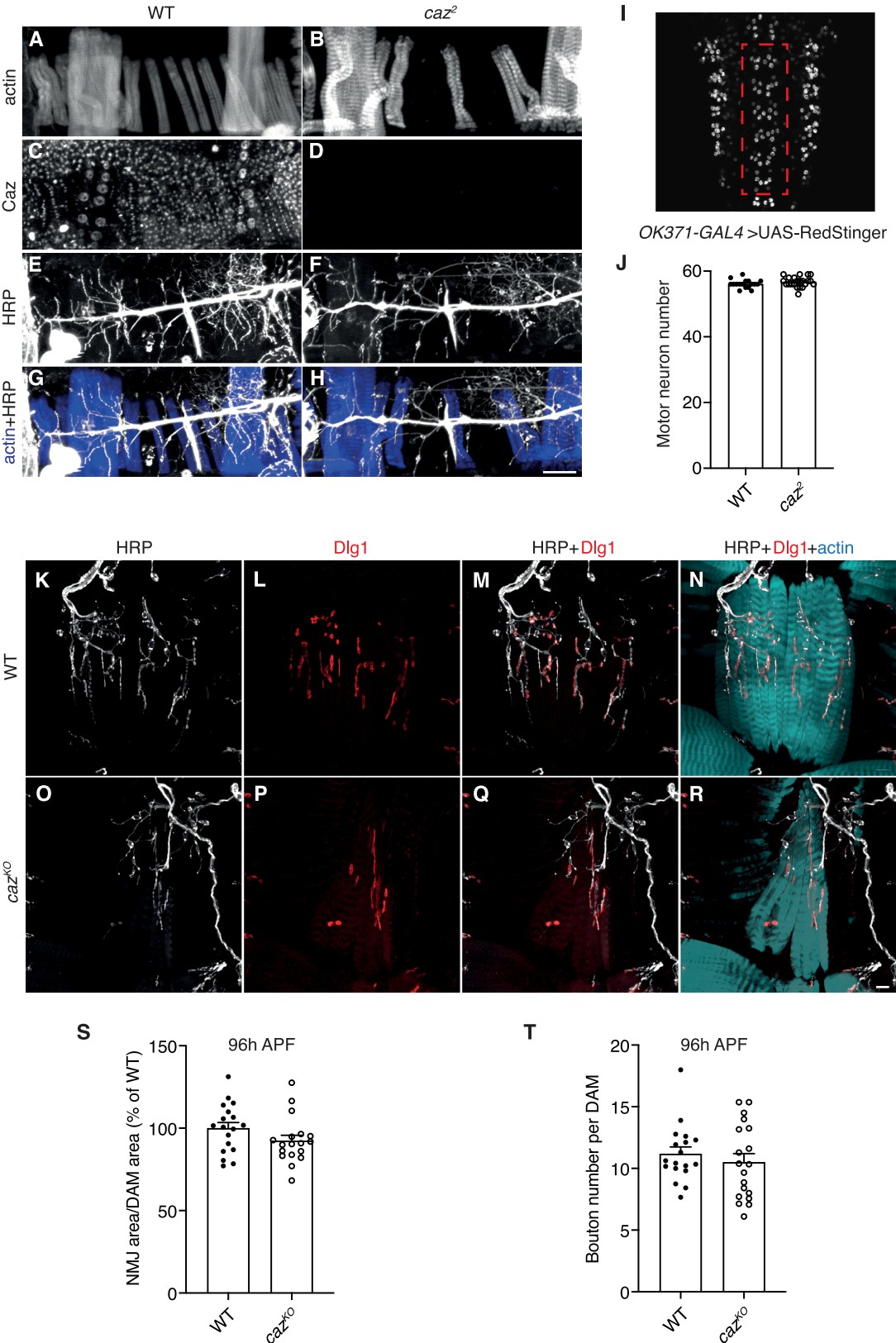

**Fig 2. Motor neuron number and muscle innervation are not affected by loss of *caz* function. A-H,** Immunostaining for actin (*A,B*), Caz (*C,D*), and HRP (*E,F*) on abdominal filets of WT (*A,C,E,G*) and *caz²* (*B,D,F,H*) pupae at 100 h APF showing

DAM morphology and innervation. The merge of actin and HRP is shown in panels G,H. Scale bar: 50μm. **I,** Central motor neuron clusters in segments A2 to A7 in the ventral nerve cord of a third instar larva are delineated by a dashed red rectangle. **J,** Quantification of motor neuron number in segments A2 to A7 in WT versus $caz^2$ third instar ventral nerve cord. Unpaired t-test; n = 17 WT versus 22 $caz^2$. Average ± SEM. **K-R,** Immunostaining for HRP (*K,O*), Dlg1 (*L,P*) and actin on abdominal filets of WT (*K-N*) and $caz^{KO}$ (*O-R*) pupae at 100 h APF showing ventral abdominal muscle morphology and innervation. Merges between HRP and Dlg1 (*M,Q*) and between HRP, Dlg1 and actin (*N,R*) are shown. Scale bar: 10μm. **S,** NMJ area relative to DAM area (shown as % of WT) in segments A3 and A4 of WT versus $caz^{KO}$ pupae at 96 h APF. Unpaired t-test; n = 18 WT versus 18 $caz^{KO}$. Average ± SEM. **T,** Bouton number per DAM in segments A3 and A4 of WT versus $caz^{KO}$ pupae at 96 h APF. Unpaired t-test; n = 18 WT versus 19 $caz^{KO}$. Average ± SEM.

## Neuron-selective loss of *caz* function causes muscle defects

The fact that (i) muscle phenotypes induced by adult myoblast-selective *caz* inactivation were not as severe as in full-body *caz* mutants, and (ii) adult myoblast-selective Caz reintroduction substantially but not fully rescued *caz* mutant muscle phenotypes, suggested that cell types other than muscle also contribute to the *caz* mutant muscle phenotypes. As *caz* function is known to be particularly important in neurons [9, 15], we evaluated whether loss of *caz* function in neurons could contribute to *caz* mutant muscle phenotypes. Consistent with this possibility, selective *caz* inactivation in neurons resulted in a significantly reduced number of founder myoblasts at 24 h APF (Fig 4A), and in a reduced DAM number and DAM misorientation at 96 h APF (Fig 4B and 4C; representative images in S3B Fig). We further evaluated the effect of neuron-selective Caz expression on *caz* mutant muscle phenotypes. Remarkably, even in a wild type genetic background, neuron-selective Caz overexpression significantly increased the number of DAMs at 96 h APF (Fig 4D). In $caz^{KO}$ pupae, neuron-selective Caz expression partially rescued the reduced DAM number (Fig 4D). Two-way ANOVA statistical analysis indicated that the DAM number was increased to a similar extent in the control and $caz^{KO}$ genetic background (the 'interaction' between *caz* genotype and Caz overexpression was not statistically significant, p = 0.086). DAM misorientation in $caz^{KO}$ pupae was substantially rescued by neuron-selective Caz expression (Fig 4E). In all, our data show that loss of *caz* function in both neurons and adult myoblasts contributes to *caz* mutant muscle phenotypes.

## Defective FGF signaling contributes to *caz* mutant muscle phenotypes

As the FGF signaling pathway has previously been implicated in adult founder myoblast selection [6], we evaluated whether defective FGF signaling might contribute to *caz* mutant muscle phenotypes. Interestingly, overexpression of the FGF receptor Heartless (Htl) in adult myoblasts (*1151-GAL4*) partially but significantly rescued the reduced DAM number and DAM misorientation in *caz* mutants (Fig 5A and 5B). Htl overexpression in an otherwise wild type background tended to increase DAM number, but this difference did not reach statistical significance. Importantly, two-way ANOVA revealed that the rescue of *caz* mutant muscle phenotypes by myoblast-selective Htl overexpression represents a genuine genetic interaction between *caz* and Htl, and is not merely an additive effect of Htl overexpression in a *caz* mutant background.

To confirm the role of the FGF pathway in *caz* mutant muscle phenotypes, we selectively overexpressed Stumps, the FGF receptor-specific signal transduction molecule downstream of Htl, in adult myoblasts. This also significantly rescued the reduced DAM number and DAM misorientation (Fig 5C and 5D) in *caz* mutants. Together these data indicate that *caz* mutant muscle phenotypes are at least in part attributable to defective FGF signaling.

These findings raised the hypothesis that Caz may, directly or indirectly, regulate the expression level of FGF pathway components. As Stumps expression becomes restricted to founder myoblasts 24 h APF [6], suggesting a key role in founder myoblast selection, we

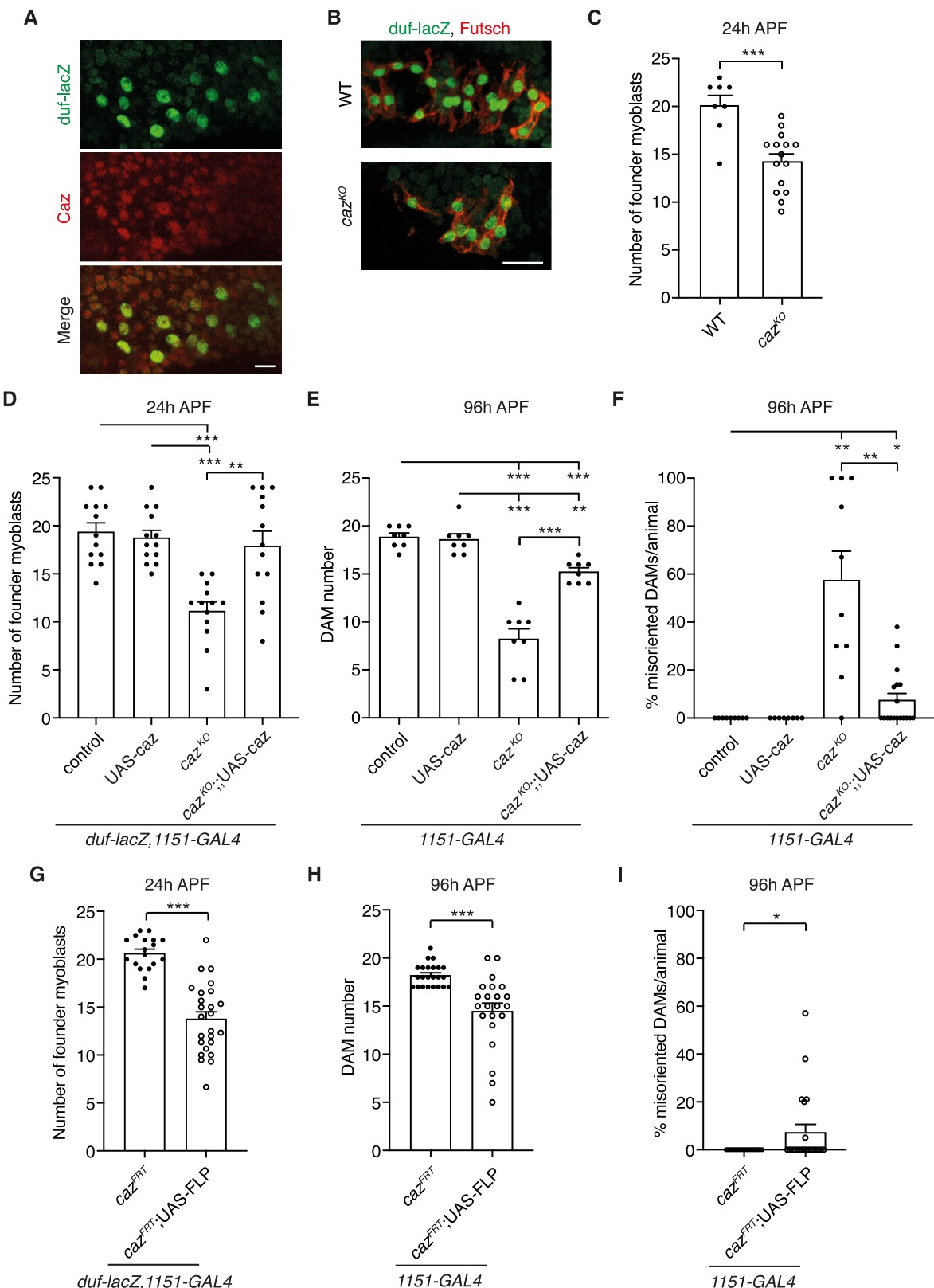

**Fig 3. Reduced founder myoblast number in *caz* mutants and rescue of *caz* mutant muscle phenotypes by adult myoblast-selective *caz* expression. A,** Immunostaining of *duf-lacZ*/Y transgenic pupal filets 24 h APF for β-galactosidase and Caz revealed Caz expression in founder myoblasts. Scale bar: 10μm. **B,** Immunostaining of founder myoblasts in a dorsal abdominal segment (A3 or A4) of 24 h APF *duf-lacZ* pupal filets, either WT (*duf-lacZ*/Y) or *caz^KO* (*caz^KO,duf-lacZ*/Y), for β-galactosidase (green) and Futsch (red). Scale bar: 20μm **C,** Quantification of the number of founder myoblasts in a dorsal abdominal segment (A3 or A4) of WT (*duf-lacZ*/Y) and *caz^KO* (*caz^KO,duf-lacZ*/Y) pupae at 24 h APF. Unpaired t-test; ***p<0.001; n = 8 WT versus 15 *caz^KO*. Average ± SEM. **D,** Quantification of the number of founder myoblasts in a dorsal abdominal segment (A3 or A4) of 24 h APF *caz^KO* pupae in which caz is selectively expressed in adult myoblasts (*caz^KO,duf-lacZ,1151-GAL4*/Y;; UAS-caz/+) as compared to the relevant control genotypes (*duf-lacZ,1151-GAL4*/Y // *duf-lacZ,1151-GAL4*/Y;; UAS-caz/+ // *caz^KO,duf-lacZ,1151-GAL4*/Y). Two-way ANOVA with Tukey's multiple comparisons test; **p<0.001, ***p<0.001; n = 13 per genotype. Average ± SEM. **E,F,** Quantification of DAM number (E) and percentage of misoriented DAMs per animal (F) in segment A4 of 96 h APF *caz^KO* pupae in which caz is selectively expressed in adult myoblasts (*caz^KO,1151-GAL4*/Y; *Mhc-GFP,his-RFP*/+; UAS-caz/+) as compared to the relevant control genotypes (*1151-GAL4*/Y; *Mhc-GFP,his-RFP*/+ // *1151-GAL4*/Y; *Mhc-GFP,his-RFP*/+; UAS-caz/+ // *caz^KO,1151-GAL4*/Y; *Mhc-GFP,his-RFP*/+). Statistics (E): Brown-Forsythe and Welch ANOVA with Dunnett's post test; **p<0.01, ***p<0.001; n = 8 per genotype. Average ± SEM. Statistics (F): One sample Wilcoxon signed rank test to compare all genotypes to control (set to 0), t-test with Welch's correction to compare *caz^KO* to *caz^KO*;; UAS-caz/+; *p<0.05, **p<0.005, ***p<0.0001; n = 9 control, 8 UAS-caz, 10 *caz^KO*, 18 *caz^KO*; UAS-caz. **G,** Quantification of the number of founder myoblasts in a dorsal abdominal segment (A3 or A4) of 24 h APF pupae in which *caz* is selectively inactivated in myoblasts (*caz^FRT,duf-lacZ,1151-GAL4*/Y; UAS-FLP/+) as compared to the relevant control genotype (*caz^FRT,duf-lacZ,1151-GAL4*/Y). Unpaired t-test; ***p<0.0001; n = 18 *caz^FRT*, 25 *caz^FRT*; UAS-FLP. Average ± SEM. **H,I,** Quantification of DAM number (H) and percentage of misoriented DAMs per animal (I) in segment A4 of 96 h APF pupae in which *caz* is selectively inactivated in myoblasts (*caz^FRT,1151-GAL4*/Y; *Mhc-GFP,his-RFP*/UAS-FLP) as compared to the relevant control genotype (*caz^FRT,1151-GAL4*/Y; *Mhc-GFP,his-RFP*/+). Statistics (H): Mann Whitney test; ***p<0.0001; n = 23 *caz^FRT*, 22 *caz^FRT*; UAS-FLP. Average ± SEM. Statistics (I): One sample Wilcoxon test; *p<0.05; n = 23 *caz^FRT*, 22 *caz^FRT*; UAS-FLP. Average ± SEM.

evaluated Stumps expression levels in adult founder myoblasts by immunostaining, using a previously described specific anti-Stumps antibody [16]. This experiment revealed significantly reduced Stumps protein levels in *caz^KO* founder myoblasts (Fig 5E and 5F). Thus, loss of *caz* function leads to reduced *Stumps* expression, which may explain the reduced founder myoblast number in *caz* mutant animals.

## Increased Xrp1 levels mediate *caz* mutant muscle phenotypes

We have previously reported that neuronal dysfunction in *caz* mutants is mediated by increased levels of Xrp1, a DNA-binding protein involved in gene expression regulation [10]. Upregulation of Xrp1 by ~3-fold was not only found in *caz* mutant nervous system but also in larval body wall, which predominantly consists of muscles [10]. We therefore evaluated whether the newly discovered *caz* mutant muscle phenotypes are also mediated by increased Xrp1 levels. Consistent with increased Xrp1 expression as the key mediator of *caz* mutant phenotypes, heterozygosity for *Xrp1* almost fully rescued the reduced DAM number (Fig 6A) and fully rescued DAM misorientation (S4A Fig) in *caz* mutant animals. Importantly, selective overexpression of Xrp1 in adult myoblasts (*1151-GAL4*) was sufficient to induce a significant reduction of DAM number (Fig 6B), as well as DAM misorientation (S4B Fig), and a substantial reduction in the number of founder myoblasts (Fig 6C). Thus, adult myoblast-selective Xrp1 overexpression phenocopied *caz* mutant muscle phenotypes. This was dependent on the DNA-binding capacity of Xrp1, as adult myoblast-selective overexpression of Xrp1 with a subtle mutation in its AT-hook DNA-binding domain did not induce muscle phenotypes (Fig 6B, S4B Fig). These data suggest that gene expression dysregulation induced by increased Xrp1 expression in adult myoblasts mediates *caz* mutant muscle phenotypes. Based on our finding that Stumps expression is reduced in *caz* mutant founder myoblasts (Fig 5E and 5F), we hypothesized that increased Xrp1 expression may suppress Stumps expression in adult founder myoblasts. Immunostaining for Stumps confirmed that Xrp1 overexpression in adult myoblasts of otherwise wild type animals was sufficient to significantly reduce Stumps levels in founder myoblasts (Fig 6D and 6E).

In addition to increased Xrp1 expression in adult myoblasts, increased Xrp1 levels in neurons may also contribute to *caz* mutant muscle phenotypes. Indeed, selective knock-down of

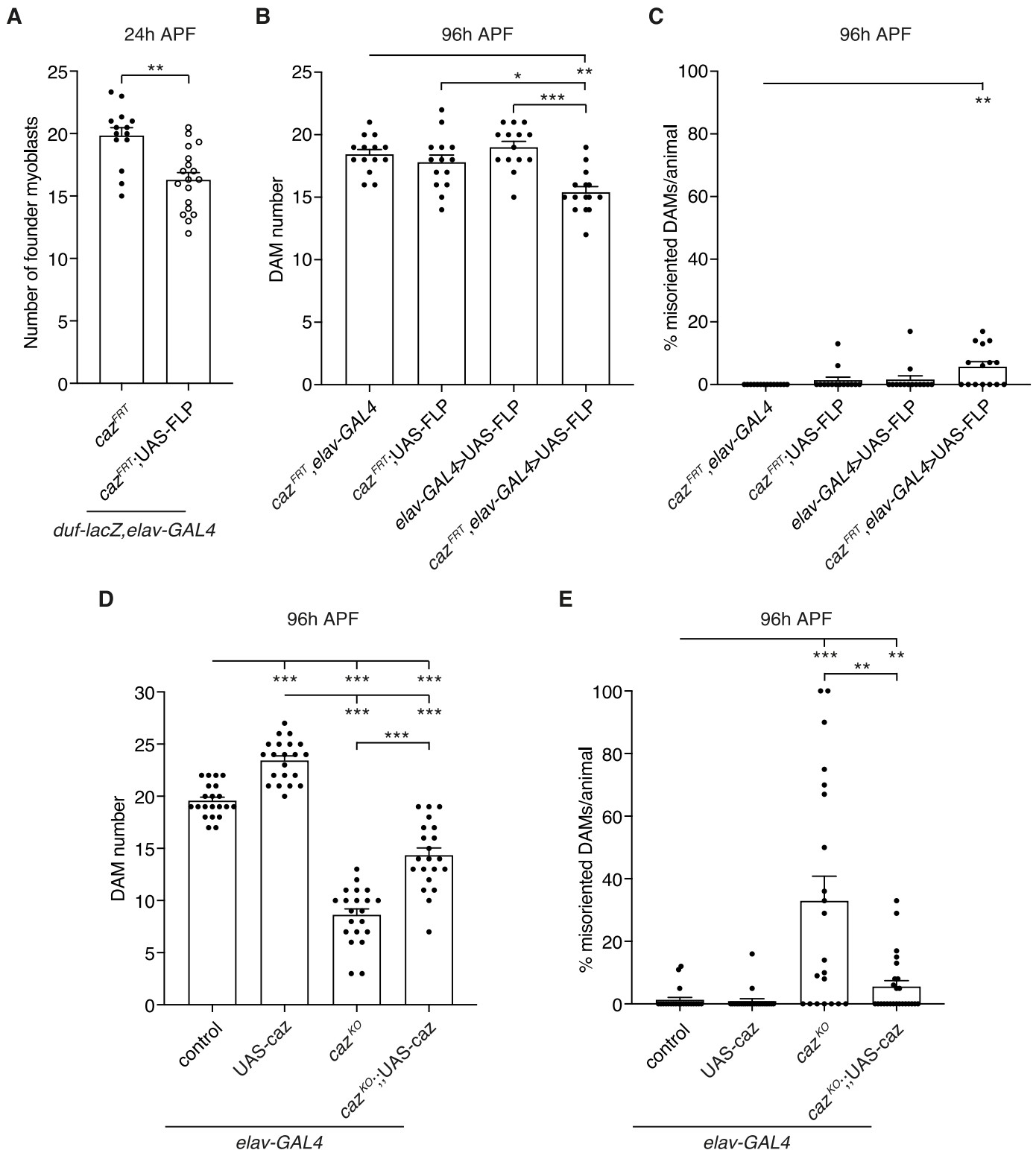

**Fig 4. Neuron-selective loss of *caz* function causes muscle defects. A,** Quantification of the number of founder myoblasts in a dorsal abdominal segment (A3 or A4) of 24 h APF pupae in which *caz* is selectively inactivated in neurons (*caz^FRT^,duf-lacZ,elav-GAL4*/Y; UAS-FLP/+) as compared to the relevant control (*caz^FRT^,duf-lacZ,elav-GAL4*/Y). Unpaired t-test; **p<0.001, n = 14 *caz^FRT^* versus n = 18 *caz^FRT^*; UAS-FLP. Average ± SEM. **B,C,** Quantification of DAM number (B) and percentage of misoriented DAMs per animal (C) in segment A4 of 96 h APF pupae in which *caz* is selectively inactivated in neurons (*caz^FRT^,elav-GAL4*/Y; *Mhc-GFP,his-RFP*/UAS-FLP) as compared to the relevant control genotypes (*caz^FRT^,elav-GAL4*/Y; *Mhc-GFP,his-RFP*/+ // *caz^FRT^*/Y; *Mhc-GFP,his-RFP*/UAS-FLP // *elav-GAL4*/Y; *Mhc-GFP,his-RFP*/UAS-FLP). Statistics (B): Kruskal-Wallis test with Dunn's multiple comparisons test; *p<0.05, **p<0.005, ***p<0.0001. Statistics (C): one sample Wilcoxon signed rank test; **p<0.01; n = 14 *caz^FRT^, elav-GAL4*, 14 *caz^FRT^*; UAS-FLP, 14 *elav-GAL4*>UAS-FLP, 15 *caz^FRT^, elav-GAL4*>UAS-FLP. Average ± SEM. **D,E,** Quantification of DAM number (D) and percentage of misoriented DAMs per animal (E) in segment A4 of 96 h APF *caz^KO^* pupae in which caz is selectively expressed in neurons (*caz^KO^,elav-GAL4*/Y; *Mhc-GFP,his-RFP*/+; UAS-caz/+) as compared to the relevant control genotypes (*elav-GAL4*/Y; *Mhc-GFP,his-RFP*/+ // *elav-GAL4*/Y; *Mhc-GFP,his-RFP*/+; UAS-caz/+ // *caz^KO^,elav-GAL4*/Y; *Mhc-GFP,his-RFP*/+). Statistics (D): two-way ANOVA with Tukey's multiple comparisons test; ***p<0.0001; n = 21 per genotype. Statistics (E): one sample Wilcoxon signed rank test to compare all genotypes to control, Mann-Whitney test to compare *caz^KO^* to *caz^KO^*; UAS-caz; **p<0.01, ***p<0.0001; n = 21 control, 23 UAS-caz, 21 *caz^KO^*, 25 *caz^KO^*; UAS-caz. Average ± SEM.

Xrp1 in neurons (*elav-GAL4*) induced a partial but substantial rescue of *caz^KO^* muscle phenotypes, both DAM number and misorientation (Fig 6F and 6G). Thus, increased Xrp1 levels in both muscle and neurons mediate *caz* mutant muscle phenotypes.

## Discussion

*Cabeza* (*caz*) is the single *Drosophila* orthologue of the human FET protein family, which consists of FUS, EWSR1 and TAF15 [7]. Caz was previously reported to have a particularly important function in neurons. Indeed, similar to full-body *caz* loss-of-function (LOF) mutants, selective *caz* inactivation in neurons induced motor deficits resulting in failure of the majority of adult flies to eclose from the pupal case, and adult escaper flies displayed dramatic climbing deficits and reduced life span [9]. Furthermore, neuron-selective expression of Caz or human FUS was sufficient to rescue the adult eclosion and motor performance defects of *caz* LOF mutants [9, 15]. Here, we show for the first time that *caz* mutants display adult abdominal muscle defects, including a reduced number and occasional misorientation of dorsal abdominal muscles (DAMs). The reduced DAM number correlated with a reduced number of adult founder myoblasts, and our data indicate that *caz* mutant adult muscle developmental defects are to an important extent attributable to loss of *caz* function in adult myoblasts. Indeed, similar to full-body *caz* LOF mutants, adult myoblast-selective inactivation of *caz* resulted in a reduced number of adult founder myoblasts and DAMs, as well as DAM misorientation. In addition, myoblast-selective Caz expression in a *caz* mutant background substantially rescued adult muscle developmental defects. Thus, *caz* function in adult muscle precursor cells is required for normal muscle development.

The normal muscle morphology and number in *caz* mutant third instar larvae indicated a selective defect in adult muscle development, with the reduced number of adult founder myoblasts as the earliest observable defect. While embryonic founder myoblast selection involves Notch-mediated lateral inhibition [5], FGF signaling through the FGF receptor Heartless (Htl) and its downstream signal transduction molecule Stumps/Dof/Hbr has been shown to mediate adult founder myoblast selection [6]. We therefore investigated the potential contribution of defective FGF signaling to *caz* mutant muscle phenotypes. We found that selective expression of Htl or Stumps in adult myoblasts significantly rescued the reduced DAM number and DAM misorientation in *caz* mutants. Consistent with this genetic evidence for a role of FGF signaling in *caz* mutant muscle phenotypes, immunostaining revealed that Stumps levels are significantly reduced in *caz* mutant founder myoblasts. Together, these data indicate that defective FGF signaling contributes to *caz* mutant muscle phenotypes.

Mallik et al. recently reported that neuronal dysfunction in *caz* loss-of-function mutants is mediated by increased expression of Xrp1 [10], a DNA-binding protein involved in protection against genotoxic stress [17], DNA damage repair [18], cell competition [19, 20] and intra-

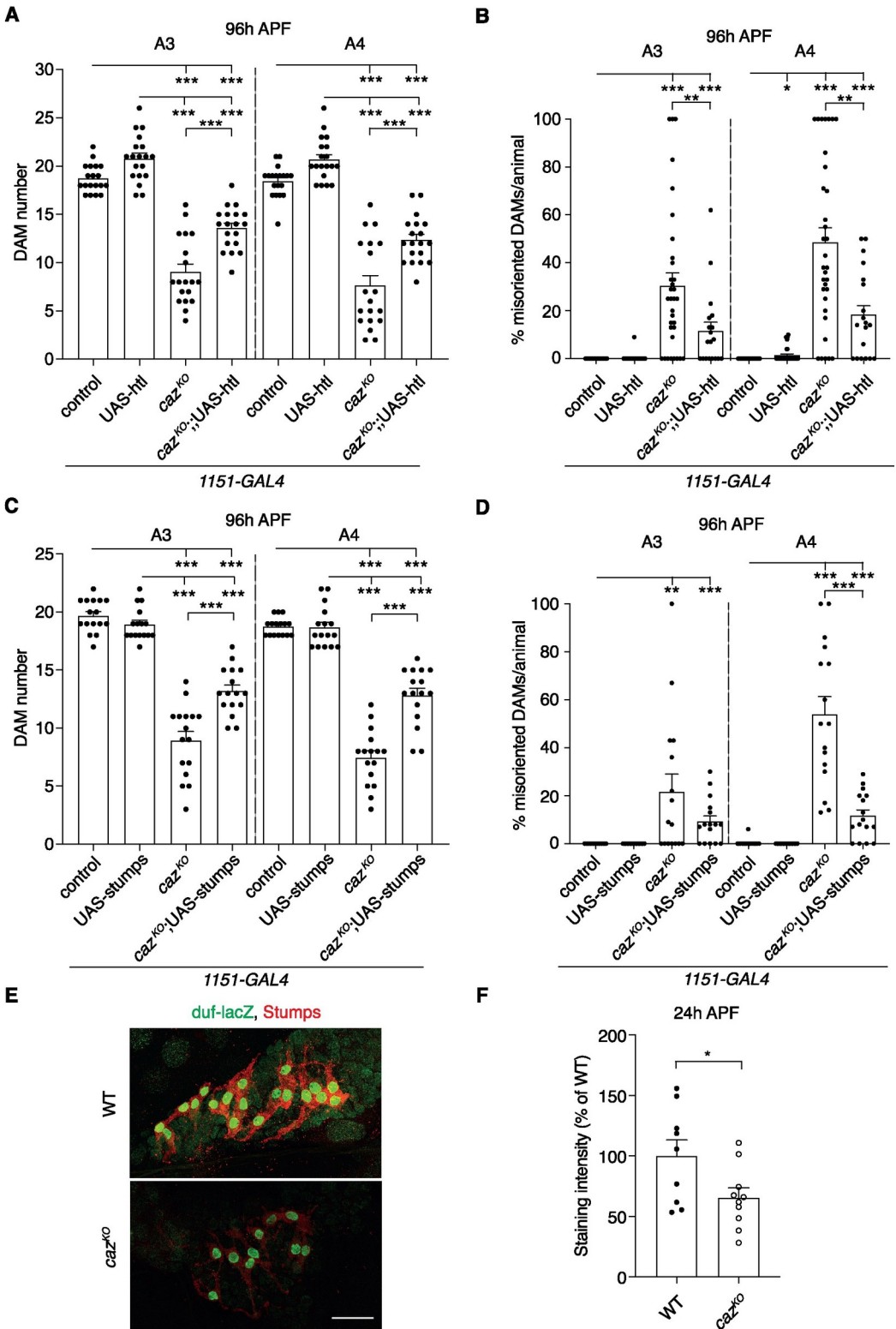

**Fig 5. Defective FGF signaling contributes to *caz* mutant muscle phenotypes. A,B,** Quantification of DAM number (A) and percentage of misoriented DAMs per animal (B) in segments A3 and A4 of 96 h APF *caz^KO* pupae in which Htl is selectively expressed in adult myoblasts (*caz^KO*,1151-GAL4/Y; Mhc-GFP,his-RFP/+; UAS-htl/+) as compared to the relevant control genotypes (*1151-GAL4/Y; Mhc-GFP,his-RFP/+ // 1151-GAL4/Y; Mhc-GFP,his-RFP/+; UAS-htl/+ // caz^KO,1151-GAL4/Y; Mhc-GFP,his-RFP/+*). Statistics (A): two-way ANOVA with Tukey's multiple comparisons test;

***p<0.0001; n = 19 per genotype. Average ± SEM. Statistics (B): one sample Wilcoxon signed rank test to compare all genotypes to control, Mann-Whitney test to compare *caz^KO* to *caz^KO*; UAS-htl; *p<0.05, **p<0.01, ***p<0.001; n = 32 control, 37 UAS-htl, 33 *caz^KO*, 19 *caz^KO*; UAS-htl. Average ± SEM. **C,D,** Quantification of DAM number (C) and percentage of misoriented DAMs per animal (D) in segments A3 and A4 of 96 h APF *caz^KO* pupae in which Stumps is selectively expressed in adult myoblasts (*caz^KO*,1151-GAL4/Y; *Mhc-GFP,his-RFP*/UAS-stumps) as compared to the relevant control genotypes (1151-GAL4/Y; *Mhc-GFP,his-RFP*/UAS-stumps/+ // *caz^KO*,1151-GAL4/Y; *Mhc-GFP,his-RFP*/+). Statistics (C): two-way ANOVA with Tukey's multiple comparisons test; ***p<0.0001; n = 16 per genotype. Average ± SEM. Statistics (D): one sample Wilcoxon signed rank test to compare all genotypes to control, Mann-Whitney test to compare *caz^KO* to *caz^KO*; UAS-stumps for A3, t-test with Welch's correction for A4; **p<0.005, ***p<0.001; n = 16 per genotype. Average ± SEM. **E,** Immunostaining of founder myoblasts in a dorsal abdominal segment (A3 or A4) of 24 h APF *duf-lacZ/Y* pupal filets, either WT (*duf-lacZ/Y*) or *caz^KO* (*caz^KO*,*duf-lacZ/Y*), for β-galactosidase (green) and Stumps (red). Scale bar: 50μm. **F,** Quantification of Stumps staining intensity (% of WT) in founder myoblasts of *caz^KO* (*caz^KO*,*duf-lacZ/Y*) 24 h APF pupae relative to WT (*duf-lacZ/Y*). Unpaired t-test; *p<0.05; n = 9 WT, 10 *caz^KO*. Average ± SEM.

and inter-organ growth coordination [21]. Increased Xrp1 expression in *caz* mutant central nervous system led to gene expression dysregulation and neuronal dysfunction [10]. In this study, we show that increased Xrp1 levels also mediate the newly-discovered *caz* mutant muscle phenotypes, as heterozygosity for *Xrp1* almost completely rescued *caz* mutant muscle phenotypes. Interestingly, selective overexpression of Xrp1 in adult myoblasts of otherwise WT animals was sufficient to induce a substantial reduction in the number of founder myoblasts and DAMs, and DAM misorientation. A subtle mutation in the Xrp1 AT-hook motif that abolishes its DNA-binding capacity completely prevented the induction of muscle phenotypes upon selective overexpression in adult myoblasts. This indicates that gene expression dysregulation as a consequence of increased Xrp1 levels mediates *caz* mutant muscle phenotypes. Intriguingly, in adult founder myoblasts, Stumps is a downstream target of Xrp1, as selective overexpression of Xrp1 in adult myoblasts of otherwise WT animals significantly decreased Stumps expression in founder myoblasts, as is the case in *caz* mutants. As Stumps expression becomes restricted to adult founder myoblasts at the time of founder myoblast selection (24 h APF) [6], suppression of Stumps by Xrp1 is likely a key molecular mechanism underlying the reduced adult founder myoblast number in *caz* mutants. Altogether, our data indicate that in adult founder myoblasts, loss of *caz* function leads to increased Xrp1 expression levels, which in turn suppresses Stumps expression, leading to a reduced number of founder myoblasts and consequently to a reduced DAM number.

Although our data indicate a role of *caz* in founder myoblast selection by controlling Xrp1 expression, which in turn regulates Stumps expression (and possibly other FGF pathway components), additional mechanisms may contribute to *caz* mutant muscle phenotypes. It is unlikely that defective myoblast fusion contributes to *caz* mutant muscle phenotypes, as the number of nuclei per DAM was not substantially changed in *caz* mutants at 96 h APF (S5 Fig). However, other processes that occur later during adult muscle development may be defective in *caz* mutants, including myoblast migration, attachment to tendons, or muscle degeneration may occur. Such mechanisms may underlie the frequent misorientation of DAMs that we observed in *caz* mutants, the mechanism of which remains to be elucidated.

Consistent with the previously recognized importance of *caz* function in neurons [9, 15], loss of caz function in neurons also contributed to *caz* mutant muscle phenotypes, despite the fact that the number of motor neuron cell bodies was not altered and that the remaining muscles in *caz* mutant animals were normally innervated. Indeed, neuron-selective inactivation of *caz* in an otherwise WT genetic background was sufficient to induce a reduced founder myoblast and DAM number, as well as DAM misorientation. Furthermore, neuron-selective reintroduction of Caz in a *caz* mutant background partially rescued the reduced DAM number

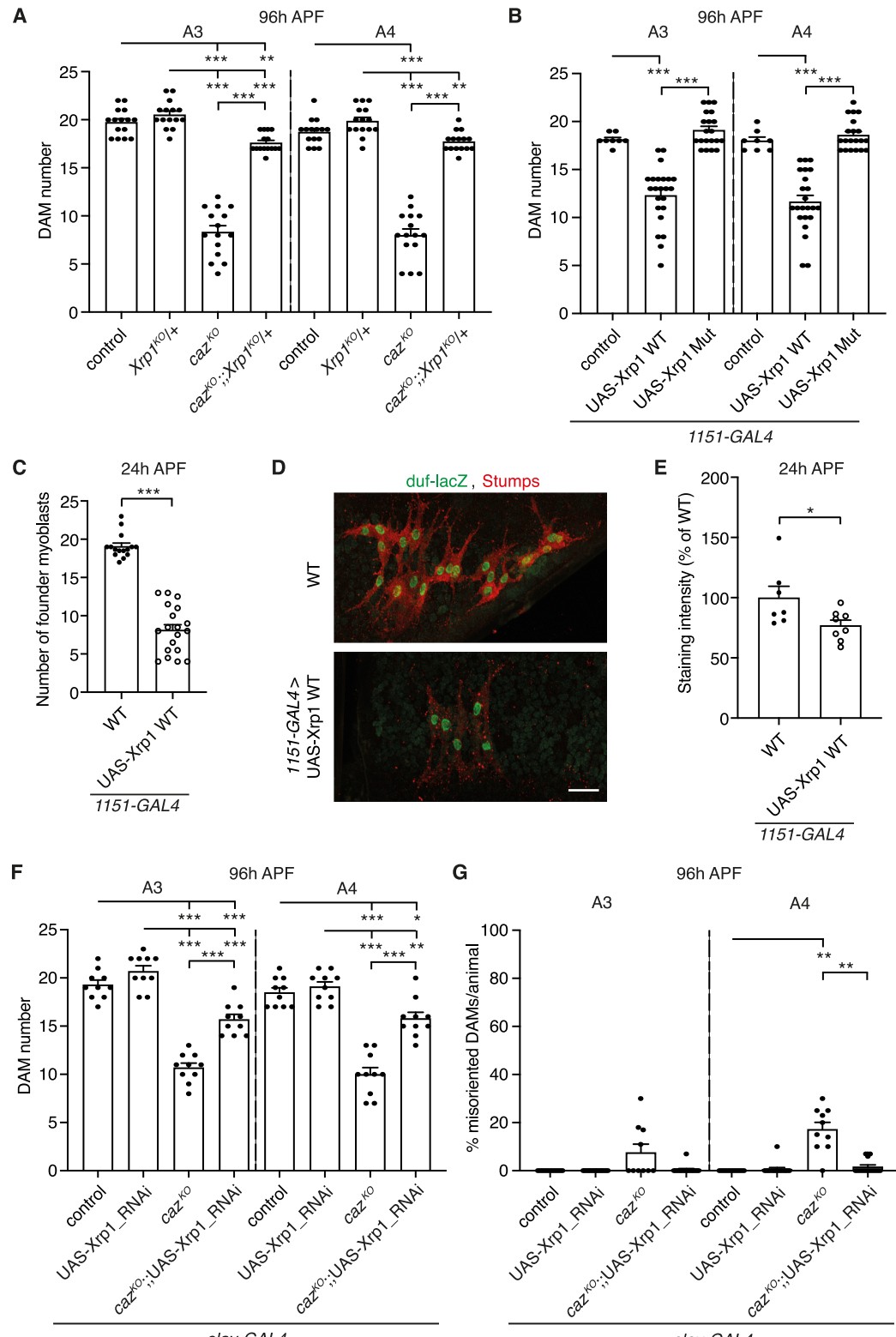

**Fig 6. Increased Xrp1 levels mediate *caz* mutant muscle phenotypes. A,** Quantification of DAM number in segments A3 and A4 of 96 h APF *caz*[KO] pupae that are heterozygous for *Xrp1*[KO] (*caz*[KO]/Y; *Mhc-GFP,his-RFP*/+; *Xrp1*[KO]/+) as compared to the relevant control genotypes (+/Y; *Mhc-GFP,his-RFP*/+ // +/Y; *Mhc-GFP,his-RFP*/+; *Xrp1*[KO]/+ // *caz*[KO]/Y; *Mhc-GFP,his-RFP*/+). Two-way

ANOVA with Tukey's multiple comparisons test; **p<0.01, ***p<0.0001; n = 15 per genotype. Average ± SEM. **B,** Quantification of DAM number in segments A3 and A4 of 96 h APF male pupae in which Xrp1 is selectively overexpressed in adult myoblasts (*1151-GAL4*), either as WT protein or with a subtle mutation that disrupts the DNA-binding capacity of the AT-hook motif (Mut), as compared to driver-only control. Brown-Forsythe and Welch ANOVA with Dunnett's post test; ***p<0.001; n = 8 control, 23 UAS-Xrp1 WT, 20 UAS-Xrp1 Mut. Average ± SEM. **C,** Quantification of the number of founder myoblasts in 24 h APF male pupae in which Xrp1 is overexpressed in adult myoblasts (*1151-GAL4*). Mann Whitney test; ***p<0.001; n = 15 WT versus 19 UAS-Xrp1 WT. Average ± SEM. **D,** Immunostaining of founder myoblasts in a dorsal abdominal segment (A3 or A4) of 24 h APF *duf-lacZ* pupal filets, either WT (*1151-GAL4,duf-lacZ/Y*) or *1151-GAL4,duf-lacZ/Y;;UAS-Xrp1 WT/+*, for β-galactosidase (green) and Stumps (red). Scale bar: 20μm. **E,** Quantification of Stumps staining intensity (% of WT) in founder myoblasts of *1151-GAL4,duf-lacZ/Y;; UAS-Xrp1 WT/+* 24 h APF pupae relative to WT (*1151-GAL4,duf-lacZ/Y*). Unpaired t-test; *p<0.05; n = 7 WT versus 8 UAS-Xrp1 WT. Average ± SEM. **F,G,** Quantification of DAM number (F) and percentage of misoriented DAMs per animal (G) in segments A3 and A4 of 96 h APF *caz^KO* pupae in which Xrp1 is selectively knocked-down in neurons (*caz^KO,elav-GAL4/Y; Mhc-GFP,his-RFP/+; UAS-Xrp1-RNAi/+*) as compared to the relevant control genotypes (*elav-GAL4/Y; Mhc-GFP,his-RFP/+ // elav-GAL4/Y; Mhc-GFP,his-RFP/+; UAS-Xrp1-RNAi/+ // caz^KO,elav-GAL4/Y; Mhc-GFP,his-RFP/+*). Statistics (F): two-way ANOVA with Tukey's multiple comparisons test; *p<0.05, **p<0.005, ***p<0.0001; n = 10 per genotype. Statistics (G): one sample Wilcoxon signed rank test to compare all genotypes to control, and *caz^KO* to *caz^KO*; UAS-Xrp1-RNAi, **p<0.005; n = 13 control, 15 UAS-Xrp1-RNAi, 10 *caz^KO*, 16 *caz^KO*; UAS-Xrp1-RNAi. Average ± SEM.

and DAM misorientation. The neuronal contribution to *caz* mutant muscle phenotypes is also mediated by increased Xrp1 expression, as neuron-selective knock-down of Xrp1 partially rescued *caz* mutant muscle phenotypes. Thus, gene expression dysregulation in *caz* mutant neurons likely explains the non-cell-autonomous contribution to muscle phenotypes. Future studies are needed to identify the Xrp1 target genes in neurons that are responsible for *caz* mutant muscle defects. In line with the neuronal contribution to *caz* mutant muscle phenotypes, the development of adult muscles is known to be intricately linked to the development of adult innervation. Early during metamorphosis, adult myoblasts migrate along motor nerves [1] and denervation of adult abdominal hemisegments at the onset of metamorphosis results in thinner and less numerous DAMs [22]. Furthermore, innervation is absolutely required for the formation of the male specific muscles [22, 23].

As mutations in FUS cause aggressive familial forms of the motor neurodegenerative disease ALS [8, 24], our findings may be interesting in the context of ALS. Indeed, a possible contribution of the skeletal muscle to ALS pathogenesis has been suggested, but is controversial. On the one hand, selective expression of ALS-mutant SOD1 in skeletal muscle of mice resulted in progressive muscle atrophy, reduced muscle strength, alteration in the contractile apparatus, mitochondrial dysfunction, paresis and motor defects [25, 26]. On the other hand, selective reduction of mutant SOD1 expression in skeletal muscle of mutant SOD1 transgenic mice did not affect their disease course, neither did increasing muscle fiber number and diameter by follistatin expression [27, 28]. Intriguingly, a recent study reported that ALS-mutant FUS is intrinsically toxic to both motor neurons and muscle cells, and toxicity in muscle may be attributable to defective FUS-mediated transcriptional regulation of acetylcholine receptor subunit genes [29]. While this and other studies in *FUS*-ALS mouse models indicate that a gain-of-toxic-function of ALS-mutant FUS is required to cause motor neuron degeneration [30–32], a possible contribution of loss of nuclear FUS function to *FUS*-ALS pathogenesis cannot be excluded. Despite the fact that *caz* mutant muscle defects are developmental, our findings are consistent with a cell-autonomous function of the *Drosophila* FUS orthologue in muscle, and a possible role of the skeletal muscle in *FUS*-ALS pathogenesis.

Interestingly, FGF signaling has previously been implicated in ALS pathogenesis, through retrograde FGF signaling from muscle to motor neuron [33]. Beyond ALS, FGF signaling has also been implicated in spinal muscular atrophy (SMA) [34], a related motor neurodegenerative disease caused by loss of *SMN1* function [35]. FUS also directly binds to SMN and associates with SMN complexes [36–39], and FUS knock-out or cytoplasmic mislocalization of ALS-mutant FUS induces cytoplasmic SMN mislocalization and a reduced number of nuclear

SMN-containing Gems [31, 36, 38, 39]. Thus, our finding that defective FGF signaling contributes to *caz* mutant muscle phenotypes is consistent with dysregulation of the FGF pathway as a contributing factor to both *FUS*-ALS and SMA, further indicating that common molecular pathways may underlie *FUS*-ALS and SMA.

## Materials and methods

### *Drosophila* genetics

Flies were kept in a temperature-controlled incubator with 12 h on/off light cycle at 25˚C. The X chromosome-inserted *elav-GAL4* (Bloomington *Drosophila* Stock center (BDSC) stock number 458) was used for pan-neuronal expression of UAS transgenes. *1151-GAL4* was used for targeted expression in adult myoblast precursors [40] (kindly provided by K. VijayRaghavan). For motor neuron quantification, *OK371-GAL4* was used to drive UAS-RedStinger. For *in vivo* imaging, *elav-GAL4* and *1151-GAL4* drivers were recombined with $caz^{KO}$, and recombinant females were crossed to *Mhc-GFP, his-RFP*/CyO [11–13] in order to generate an introgressed stock. To induce cell-type-specific *caz* inactivation, $caz^{FRT}$ [9] was recombined with *elav-GAL4* and *1151-GAL4*; homozygous females were crossed to UAS-FLP (BDSC stock 4539) males and male offspring was used for the experiments. In order to evaluate the number of founder myoblasts at 24 h APF, *1151-GAL4* // $caz^{KO}$ // $caz^{KO}$,*1151-GAL4* // $caz^{FRT}$,*1151-GAL4* // $caz^{FRT}$,*elav-GAL4* were recombined with *duf-lacZ* [41]. The UAS-caz line used for rescue experiments was the UAS-flag-caz line described in [15]. The htl line used in this study was UAS-htl.ORF.3xHA (F000798; FlyORF). The UAS-stumps transgenic line was provided by M. Leptin [16]. The $caz^2$ and $caz^{KO}$ lines [9], as well as $Xrp1^{KO}$ and 5xUAS-Xrp1 lines were described [10] and the UAS-Xrp1-RNAi line used in this study was P[TRiP.HMS00053]attP2 (BDSC 34521).

### *In vivo* imaging of dorsal abdominal muscles

To image dorsal abdominal muscles, appropriate crosses were set up, male larvae were selected and placed in a new vial containing standard *Drosophila* food supplemented with yeast paste. White prepupae were collected on wet Whatman filter paper in petri dishes and kept at 25˚C for 96 h. After 96 h, pharate adults were dissected out from the pupal case and placed in an approximately 2 mm furrow of a plastic slide with a drop of 60% glycerol. Images of serial optical sections were acquired using a Zeiss LSM700 or a Leica SP8 laser scanning confocal microscope. For quantification, DAMs were manually counted using ImageJ/Fiji [42] by inspecting the acquired Z-stacks.

Histone-RFP (*his-RFP*) expression was used to quantify the number of nuclei that are present in the DAMs in abdominal segments A3 and A4.

### Motor neuron quantification

The number of motor neuron cell bodies were quantified in third instar larval ventral nerve cords (VNC) of $caz^2$ and WT controls. Motor neuron cell bodies were visualized by *OK371-GAL4*-driven UAS-RedStinger and the number of motor neuron cell bodies was counted in the central clusters in segments A2-A7 of the 3rd instar larval VNC.

### Immunostainings

**Immunostaining of pharate adults (96 h APF).** Open book preparations of pupal abdomens were fixed in 4% PFA in PBS for 30 minutes. Tissues were washed 3 x 10 min with PBS/0.2% Triton X-100 and blocked for 1 h at RT with 10% goat serum in PBS. Tissues were

incubated overnight at 4˚C with primary antibodies against Caz (mouse monoclonal clone 3F4; 1:30; [43]) or Dlg1 (mouse monoclonal clone 4F3; 1:200; Developmental Studies Hybridoma Bank). Secondary antibodies conjugated either with Alexa Fluor 488 or Alexa Fluor 568 (1:500; Molecular Probes), anti-HRP-549 (1:1000; Jackson ImmunoResearch) and Phalloidin-647 (1:20; Cell Signaling) were applied for 3 h at RT.

For quantification of neuromuscular endplate area we used ImageJ/Fiji in order to measure the endplate and muscle area in DAMs of abdominal segment A3 and A4, from maximum intensity projections of acquired confocal z-stacks. In case of multiple branches, the sum of the areas of individual branches was used as total endplate area. The endplate areas were then normalized to the respective muscle area.

**Immunostaining of 24 h APF pupae.** Open book preparations of abdomens were fixed in 4% PFA in PBS for 60 minutes. Tissues were washed 3 x 10 min with PBS/0.2% Triton X-100 and blocked for 1 h at RT in 10% goat serum in PBS.

For quantification of founder myoblasts and Caz immunostaining, tissues were incubated overnight at 4˚C with the first primary antibody against β-galactosidase (mouse monoclonal 40-1a; 1:50; Developmental Studies Hybridoma Bank). Anti-mouse secondary antibody conjugated with Alexa Fluor 488 (1:500; Molecular Probes) was applied at RT for 1 h. Tissues were then washed 3 x 10 min in PBS/0.2% Triton X-100 and incubated overnight at 4˚C with the second primary antibody against Caz or Futsch (mouse monoclonal clone 22C10; 1:100; Developmental Studies Hybridoma Bank). Anti-mouse secondary antibody conjugated with Alexa Fluor 568 (1:250; Molecular Probes) was applied at RT for 30 min. For imaging a Zeiss LSM700 or a Leica SP8 laser scanning confocal microscope was used.

For quantification of Stumps level, tissues were incubated overnight at 4˚C with primary antibodies against β-galactosidase (mouse monoclonal 40-1a; 1:50; Developmental Studies Hybridoma Bank) and Stumps, a gift from M. Leptin (rabbit; 1:1000) [16]. Secondary antibodies conjugated either with Alexa Fluor 488 or Alexa Fluor 568 (1:500; Molecular Probes) were applied at RT for 1 h. As internal control, TetraSpeck Microspheres 4.0 μm (ThermoFisher) were applied on the slides while mounting the tissues. Images were acquired using a Leica SP8 laser scanning confocal microscope with a 40x/1.3 NA oil objective with the exact same settings for all the samples. Maximum intensity projections of acquired z-stacks were generated with ImageJ/Fiji. The mean cytoplasmic intensity was measured in these projections using a custom-made ImageJ/Fiji macro, upon thresholding and removing all background pixels' intensities. Same approach was used for the quantification of beads' intensity. For both Stumps staining and internal control, the mean intensity per pupa was calculated and the cytoplasmic Stumps staining intensity was normalized to the beads intensity.

## Larval muscle quantification

To visualize larval muscles, *elav-GAL4*; *Mhc-GFP* and *caz^KO*; *Mhc-GFP* larvae were dissected and briefly fixed in Bouin's solution for 3 min, followed by 3 x 15-min washes with PBS/0.2% Triton X-100 and three 15-min washes in PBS, and mounted in Vectashield (Vector Laboratories). Samples were imaged using a Leica SP8 laser scanning confocal microscope. For quantification, larval muscles in A3 and A4 were counted using ImageJ/Fiji.

## Statistical analysis

All results from analysis are presented as mean ± standard error of the mean (SEM) and differences were considered significant when p <0.05. Before analysis, a Robust regression and Outlier removal method (ROUT) was performed to detect all outliers. This nonlinear regression method fits a curve that is not influenced by outliers. Residuals of this fit were then analyzed

by a test adapted from the False Discovery Rate approach, to identify any outliers. This analysis did not identify outliers, and therefore no data points were removed. Normality and homoscedasticity of all data was analyzed by a Shapiro–Wilk and Brown–Forsythe (F-test for t-tests) test, respectively. Subsequent statistical tests were only performed if all assumptions were met, except mild heteroscedasticity that was observed between experimental groups in Figs 3F, 5A, 5B, 6A, S3A and S3B(A3).

For comparison of normally distributed data of two groups, two-tailed unpaired Student's *t*-test was used in combination with an *F*-test to confirm that the variances between groups were not significantly different. Non-parametric Mann–Whitney tests were performed if data were not normally distributed. An unpaired *t*-test with Welch's correction was performed for data with heteroscedasticity.

Comparisons of data consisting of more than two groups, varying in a single factor, were performed using Kruskal–Wallis for not-normally distributed data with homogeneous variance, and using Brown-Forsythe with Welch's correction for normally distributed data with heterogeneous variance, both using Dunn's or Dunnett's multiple comparisons post hoc test.

Comparisons of data consisting of more than two groups, varying in two factors, was performed using two-way ANOVA and subsequent Tukey's multiple comparisons test for normally distributed data with homogeneous variance.

To analyze muscle misorientation, one sample two-tailed Wilcoxon signed rank test using the Pratt method was used to evaluate differences as compared to the control genotype (with a set value of 0).

Data were analyzed using GraphPad Prism v.8.0 and R (3.5.1).

## Supporting information

**S1 Fig. Reduced DAM number in both age-matched and stage-matched *caz*$^{KO}$ pupae. A,B,** Quantification of DAM number in segments A3 and A4 of either age-matched (100 h APF, *A*) or stage-matched (P15: twitching legs, *B*) *caz*$^{KO}$ versus WT male pupal filets immunostained for actin. Unpaired t-test (*A*, *B*-A3) and t-test with Welch's correction (*B*-A4); **p<0.01, ***p<0.001; n(A) = 4 WT versus 5 *caz*$^{KO}$ (A3), 5 WT versus 4 *caz*$^{KO}$ (A4); n(B) = 11 WT versus 10 *caz*$^{KO}$ (A3), 15 WT versus 8 *caz*$^{KO}$ (A4). Average ± SEM.
(TIF)

**S2 Fig. Loss of *caz* function does not affect larval muscle morphology. A,** A transgenic line in which the *Myosin heavy chain* gene is GFP-tagged (*Mhc-GFP*) allowed for visualization of muscles in abdominal segments A3 and A4 of WT (+/Y; *Mhc-GFP,his-RFP*/+) and *caz*$^{KO}$ (*caz*$^{KO}$/Y; *Mhc-GFP,his-RFP*/+) third instar larvae. Scale bar: 100μm. **B,** Quantification of larval muscle number in segments A3 and A4. Mann-Whitney test; n = 6 per genotype. Average ± SEM.
(TIF)

**S3 Fig. DAM phenotypes induced by selective *caz* inactivation in adult myoblasts or neurons. A,** Representative images of DAMs in abdominal segments A3 and A4 of 96 h APF pupae in which *caz* was selectively inactivated in adult myoblasts (*caz*$^{FRT}$,*1151-GAL4*/Y; *Mhc-GFP*/UAS-FLP, right panel) as compared to the relevant control (*caz*$^{FRT}$,*1151-GAL4*/Y; *Mhc-GFP*/+, left panel). Scale bar: 100μm. **B,** Representative images of DAMs in abdominal segments A3 and A4 of 96 h APF pupae in which *caz* was selectively inactivated in neurons (*caz*$^{FRT}$,*elav-GAL4*/Y; *Mhc-GFP*/UAS-FLP, right panel) as compared to the relevant control (*caz*$^{FRT}$,*elav-GAL4*/Y; *Mhc-GFP*/+, left panel). Scale bar: 100μm.
(TIF)

**S4 Fig. Increased Xrp1 levels mediate *caz* mutant DAM misorientation. A,** Percentage of misoriented DAMs per animal in segments A3 and A4 of 96 h APF *caz^{KO}* pupae that are heterozygous for *Xrp1* (*caz^{KO}*/Y; *Mhc-GFP,his-RFP*/+; *Xrp1^{KO}*/+) as compared to the relevant control genotypes (+/Y; *Mhc-GFP,his-RFP*/+ // +/Y; *Mhc-GFP,his-RFP*/+; *Xrp1^{KO}*/+ // *caz^{KO}*/Y; *Mhc-GFP,his-RFP*/+). One sample Wilcoxon signed rank test to compare all genotypes to control, and *caz^{KO}* to *caz^{KO}*; *Xrp1^{KO}*/+; ***p<0.0005; n = 15 per genotype. Average ± SEM. **B,** Percentage of misoriented DAMs per animal in segments A3 and A4 of 96 h APF male pupae that selectively overexpress Xrp1 in adult myoblasts (*1151-GAL4*), either as WT protein or with a subtle mutation that disrupts the DNA-binding capacity of the AT-hook motif (Mut), as compared to driver-only control. One sample Wilcoxon signed rank test to compare all genotypes to control and UAS-Xrp1 WT to UAS-Xrp1 Mut; **p<0.01, ***p<0.0001; n = 8 control, 23 UAS-Xrp1 WT, 20 UAS-Xrp1 Mut. Average ± SEM.
(TIF)

**S5 Fig. Loss of *caz* function does not preclude myoblast fusion.** The average number of nuclei per DAM was determined in abdominal segments A3 and A4 of 96 h APF WT, *caz²*, and *caz^{KO}* pupae that carried *Mhc-GFP* and *his-RFP* transgenes to visualize muscles and nuclei, respectively. Ordinary one-way ANOVA with Dunnett's post test; *p<0.05; n = 7 per genotype. Average ± SEM.
(TIF)

## Acknowledgments

We thank K. VijayRaghavan, M. Leptin and B. McCabe for providing fly lines, M. Leptin for providing the Stumps antibody, and the General Instruments Department of the Faculty of Science of Radboud University. We thank P. Callaerts and A.F. Siekmann for constructive feedback on this manuscript.

## Author Contributions

**Conceptualization:** Marica Catinozzi, Marie Frickenhaus, Erik Storkebaum.

**Formal analysis:** Marica Catinozzi, Moushami Mallik, Marie Frickenhaus, Marije Been, Divita Kulshrestha, Erik Storkebaum.

**Funding acquisition:** Erik Storkebaum.

**Investigation:** Marica Catinozzi, Moushami Mallik, Marie Frickenhaus, Marije Been, Céline Sijlmans, Divita Kulshrestha, Manuela Weitkunat.

**Methodology:** Ioannis Alexopoulos.

**Project administration:** Erik Storkebaum.

**Resources:** Frank Schnorrer.

**Supervision:** Erik Storkebaum.

**Writing – original draft:** Marica Catinozzi, Erik Storkebaum.

**Writing – review & editing:** Frank Schnorrer.

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
