## [Decision Letter · Decision Letter 0]

1 Nov 2019

Dear Dr Storkebaum,

Thank you very much for submitting your Research Article entitled 'The Drosophila FUS ortholog caz directs adult founder myoblast selection by Xrp1-dependent regulation of FGF signaling' to PLOS Genetics. Your manuscript was fully evaluated at the editorial level and by two independent peer reviewers. The reviewers thought that you had addressed an important question, but both raised some substantial concerns about what they consider as inconsistencies and overstatements that need to be amended. Based on the reviews, we will not be able to accept this version of the manuscript, but we would be willing to review again a much-revised version. We cannot, of course, promise publication at that time.

If you decide to revise the manuscript for further consideration at PLOS Genetics, please aim to resubmit within the next 60 days, unless it will take extra time to address the concerns of the reviewers, in which case we would appreciate an expected resubmission date by email to plosgenetics@plos.org.

[LINK]

We are sorry that we cannot be more positive about your manuscript at this stage. Please do not hesitate to contact us if you have any concerns or questions.

Yours sincerely,

Claude Desplan

Associate Editor

PLOS Genetics

Gregory P. Copenhaver

Editor-in-Chief

PLOS Genetics

Reviewer's Responses to Questions

**Comments to the Authors:**

Reviewer #1: The paper by Catinozzi et al. focuses on the contribution of the Drosophila FUS ortholog “Cabeza” (caz) to adult founder myoblast selection by Xrp-dependent regulation of FGF signaling. Caz appears to function in both motor neurons, as well as in dorsal abdominal muscles (DAMs) to promote proper DAM pattern and number. The authors investigated the role of cab in muscles, and discovered that similarly to the genetic connections described for cab in neurons, cab is acting upstream of the FGF receptor pathway and controls the levels of its downstream effector Stumps in the DAM founder cells. In cab mutants the levels of Stumps in these cells are lower resulting with reduced number of muscles, as well as with DAMs exhibiting aberrant pattern. Overexpression of Stumps or the FGF receptor Htl in myoblasts partially rescued these phenotype. Further experiments indicated that the caz phenotype is induced due to overexpression of Xrp1, a DNA binding protein involved in gene expression regulation, and that rescue of caz can be obtained by combining caz mutants with heterozygous Xrp1 alleles. In addition, a similar DAM phenotype (observed in caz) can be obtained by overexpression of Xrp1 in a wild type myoblasts, supporting the idea that caz represses Xrp1 levels. These results are interesting, the genetic experiments are very convincing and the images and statistics are appropriate.

1. A major concern lies in the conclusion of the authors regarding the specific activity of caz in muscle founder cells. FGF signaling is presumably required not only in founder cells but also in the myoblasts, for their migration, guidance, fusion and attachment. The broad expression of caz, and in particular the fact that the authors used for all their rescue experiments a GAL4 driver which is expressed in all myoblasts (and not a founder cell specific Gal4 driver) do not support a specific function of caz in founder cells selection, but rather in various FGF-dependent activities in the myoblasts.

2. Fig 2 – how the authors prove that the innervation pattern is normal? In my view the innervation pattern is different from that of wild type. It might be due to the aberrant muscle pattern, but this should be better analyzed and quantified.

3. Fig 3B -Duf-lacZ might persist longer than the actual protein, and it appears that the image in Fig 3B shows myotubes rather than founder cells.

4. Discussion – is too lengthy. I think the last part discussing speculations about ALS and SMA should be shorten.

Reviewer #2: The manuscript by Catinozzi et al presents functional analysis of Drosophila FUS ortholog caz during adult muscle development. Authors demonstrate that caz acts via Xrp1 and regulates FGF signaling known to trigger adult founder myoblast specification. They analyse, in different genetic contexts, number of Duf-positive dorsal adult founders and number of derived from these founders dorsal abdominal muscles (DAMs). They conclude that caz influences DAMs development both cell autonomously and via its role in motor neurons. Authors draw then parallels between their findings and contribution of muscle defects to pathogenesis of ALS and SMA neurodegenerative diseases in which FUS and FGF are involved. Overall, this study identifies caz as a new player in adult Drosophila myogenesis and points to its role in coordinated development of adult muscles and motor neurons.

It is a valuable study but it carries are several inconsistencies and overstatements that need to be amended.

1. Authors need to clarify what are the genotypes tested in the case GAL4 drivers are used (Figs. 3, 5, 6). For example in Fig. 3D is « control » corresponding to 1151-GAL4 homozygous or to 1151-GAL4/+ ? Full genotype descriptions for all GAL4-involving contexts is required.

2. To determine contribution of adult myoblasts versus neurons to casKO phenotypes authors have to use the same genetic contexts in rescue experiments. In Fig. 3D casKO alone reduces DAM number to less than 10 wereas in Fig. 4C casKO in the context of UAS-cas leads to a milder reduction of DAM number (more than 10). This makes rescue experiments incompatible for comparison. It also indicates that UAS-cas transgene could be leaky….. Did authors test this ?

3. Number of misoriented DAMs par animal in casKO is about 10% in Fig. 1E but more than 50% in Fig.3E – this needs to be clarified !

4. Representative views of phenotypes for conditional cazKO in muscles and in motor neurons need to be shown (at least in a supplementary fig).

5. Is conditional caz knockout in motor neurons affecting number of founders ? This experiment could help in defining wherther caz expressed in motor neurons influences specification of adult founders

6. Measurement of intensity of stumps signal is not convincing. (Fig. 5C,D) Intensity of stumps signal needs to be referenced to an internal control signal that is not changing in casKO context and ideally also to the background signal. In Fig. 5C in casKO duf-lacZ seems reduced compared to wt. It is thus not a good reference.

Authors have to provide more details about how measurements of intensity was performed using ImageJ.

7. In discussion authors claim :

« ….. our data indicate that caz mutant adult muscle developmental defects are to a major extent attributable to loss of caz function in adult myoblasts. Indeed, similar to full-body caz LOF mutants, adult myoblast-selective inactivation of caz resulted in a reduced number of DAMs, as well as their misorientation…. »

This appears to be overstatement considering that conditional casKO in muscle precursors (Fig.3F) leads to loss of DAMs that is similar to that of conditional casKO in motor neurons (Fig.4A).

8. « directs » in title is not appropriate as it suggests that without caz founders are not selected. It could be replace by « promotes »

Minor points :

1.Misorientation of some of DAMs, as discussed by authors, could indicate affected capacity to interact with tendon cells, another role of caz. Testing expression of betaPS Integrin could help in defining whether misoriented DAMs keep capacities to attach.

2. Did authors test cazRNAi knockdowns with different drivers, for example in flight muscles?

**Have all data underlying the figures and results presented in the manuscript been provided?**

Reviewer #1: Yes

Reviewer #2: Yes

PLOS authors have the option to publish the peer review history of their article (what does this mean?). If published, this will include your full peer review and any attached files.

Reviewer #1: No

Reviewer #2: No

---

## [Decision Letter · Decision Letter 1]

20 Mar 2020

Dear Dr Storkebaum,

We are pleased to inform you that your manuscript entitled "The Drosophila FUS ortholog cabeza promotes adult founder myoblast selection by Xrp1-dependent regulation of FGF signaling" has been editorially accepted for publication in PLOS Genetics. Congratulations!

Yours sincerely,

Claude Desplan

Associate Editor

PLOS Genetics

Gregory P. Copenhaver

Editor-in-Chief

PLOS Genetics

Comments from the reviewers (if applicable):

Reviewer's Responses to Questions

Comments to the Authors:

Please note here if the review is uploaded as an attachment.

Reviewer #1: The authors fully addressed my comments.

Reviewer #2: Authors made several amendments and revisions in the text and in the figures, which clarified all points raised in my comments. The current version of the manuscript appears appropriate for publication in PLoS Genet

Have all data underlying the figures and results presented in the manuscript been provided?

Large-scale datasets should be made available via a public repository as described in the 

PLOS Genetics

data availability policy, and numerical data that underlies graphs or summary statistics should be provided in spreadsheet form as supporting information.

Reviewer #1: None

Reviewer #2: Yes

PLOS authors have the option to publish the peer review history of their article (what does this mean?). If published, this will include your full peer review and any attached files.

Do you want your identity to be public for this peer review?

 For information about this choice, including consent withdrawal, please see our Privacy Policy.

Reviewer #1: No

Reviewer #2: No

**Data Deposition**

http://datadryad.org/submit?journalID=pgenetics&manu=PGENETICS-D-19-01711R1

Press Queries
